# Dynamical Tunneling in More than Two Degrees of Freedom

**DOI:** 10.3390/e26040333

**Published:** 2024-04-14

**Authors:** Srihari Keshavamurthy

**Affiliations:** Department of Chemistry, Indian Institute of Technology, Kanpur 208016, Uttar Pradesh, India; srihari@iitk.ac.in

**Keywords:** dynamical tunneling, resonance-assisted tunneling, chaos-assisted tunneling, Hamiltonian systems, Arnold web, multidimensional phase space

## Abstract

Recent progress towards understanding the mechanism of dynamical tunneling in Hamiltonian systems with three or more degrees of freedom (DoF) is reviewed. In contrast to systems with two degrees of freedom, the three or more degrees of freedom case presents several challenges. Specifically, in higher-dimensional phase spaces, multiple mechanisms for classical transport have significant implications for the evolution of initial quantum states. In this review, the importance of features on the Arnold web, a signature of systems with three or more DoF, to the mechanism of resonance-assisted tunneling is illustrated using select examples. These examples represent relevant models for phenomena such as intramolecular vibrational energy redistribution in isolated molecules and the dynamics of Bose–Einstein condensates trapped in optical lattices.

## 1. Introduction

The phenomena of dynamical tunneling (DT), introduced by Davis and Heller nearly four decades ago [1] and anticipated in earlier studies, is associated with processes that are classically forbidden but occur quantum mechanically. The reader may consult ref. [2] for a detailed historical perspective on dynamical tunneling. More importantly, such processes in classical dynamics may be energetically allowed yet dynamically forbidden. Thus, while no barriers may be apparent in the configuration space, dynamical barriers can and do exist in the full phase space of the system. Quantum dynamics can then mix near-degenerate states via tunneling through such dynamical barriers. A profound consequence is the proliferation of quantum pathways that open up for the dynamical evolution of an initial quantum state. As a consequence, substantial differences between the classical and quantum dynamics can emerge in the limit of “sufficiently long” timescales.

The additional pathways afforded via DT influence several observables of interest, with relevance to a wide range of physical systems [3]. In the molecular context, examples come from high-resolution spectra wherein “clumps” of spectral lines are associated with the vibrational superexchange mechanism [4,5] and related to quantum vibrational energy flow in molecules [6,7,8]. Additional examples include the generation of NOON states in ultracold bosonic atoms [9,10], the efficacy of coherent control in driven [11,12,13,14] and kicked [15,16] systems, stability of quantum discrete breathers [17,18], and fragmentation of trapped Bose–Einstein condensates [19,20]. Furthermore, dynamical tunneling has been shown to play a crucial role in a variety of “engineered” systems [21,22,23,24,25,26,27,28,29,30,31,32,33,34].

Interestingly, although it is a purely quantum effect, it has been established that the nature of the classical phase space has a strong influence on DT. Indeed, the proposed mechanisms typically invoke specific structures in the phase space. For example, the importance of nonlinear resonances, chaos, Kolmogorov-Arnold-Moser (KAM), and partial barriers in the multidimensional phase space has been clearly established. Thus, the resonance-assisted (RAT) [35,36,37,38,39,40] and chaos-assisted tunneling (CAT) [41,42] mechanisms have been studied rather extensively in the context of Hamiltonian systems with two degrees of freedom. The former is relevant in near-integrable systems wherein near-degenerate quantum states can mix due to the presence of specific nonlinear resonances. For smaller effective *ℏ* the near-degenerate states can be coupled via a multitude of nonlinear resonances [37,39]. In the case of CAT, one invokes a coupling between the symmetric regular islands and the chaotic sea. The chaotic sea is modeled via random matrix theory with a typical three-level scenario that involves avoided crossing of the tunneling doublet with a chaotic state [41,43]. In mixed systems, one needs to invoke [36] both RAT and CAT in general for a proper quantitative description of DT. A comprehensive review of the different mechanisms and their interplay can be found in the articles in the two edited volumes [44,45]. On the other hand, Shudo and coworkers have recently suggested [46,47,48] that in the limit of “ultra” near-integrable systems, enhancements in tunneling probabilities may not correspond to any classical phase space structure and a careful look at the complex phase space dynamics is necessary. Nevertheless, in this review, the former viewpoint is taken for a couple of reasons. First, typical physical systems are far from the ultra near-integrable limit and have mixed regular–chaotic phase space. One anticipates that the extent of modulation of DT due to the phase space features will outweigh any purely quantum contribution. Second, as exemplified by the RAT and CAT studies, identifying specific phase space structures as dominant contributors allows for a predictive semiclassical theory [2,41,48].

Despite extensive investigations of DT over the past few decades, the fact remains that to date, very few studies have been performed for systems with three or more degrees of freedom [49,50,51,52]. There are several reasons for this. Chief among them has to do with the nontrivial change in the phase space topology, and hence transport, in going from f=2 to f≥3 Hamiltonians. As is well known, whereas mixed regular–chaotic phase spaces can manifest in both cases, in the f≥3 case, the chaotic regions are no longer disconnected. Specifically, the chaotic regions associated with the destroyed separatrices of the various nonlinear resonances form an intricate network known as the Arnold web. It is thus possible for “distant” regions in the phase space to be connected by purely classical transport. One such mechanism for phase space transport in the near-integrable limit is known as Arnold “diffusion”, which is expected to occur on exponentially long timescales. It should be noted here that the transport in the connected chaotic layer due to the Arnold mechanism is not necessarily a normal diffusive process over moderately long timescales. Thus, associating a diffusion constant with the process is questionable [53,54]. One may perhaps argue that the Arnold diffusion timescale is much longer than DT (or, for that matter, any physically relevant) timescale, and hence of limited interest—a sentiment already expressed by Davis and Heller [1] when they concluded their classic study by saying that “Identification of dynamical tunneling in multidimensional systems may be a matter of comparing a small flow classically to a large quantum mechanical coupling”. However, for f≥3, Arnold diffusion is not the only mechanism that leads to transport. In the nonintegrable limit, there are indications [50,55,56,57,58,59,60,61] that it is possible to have different mechanisms that lead to relatively faster exploration of the phase space. It is important to note that many of these mechanisms are operative only for systems with f≥3 since they require the connected Arnold web structure. One key example involves a feature on the Arnold web known as a resonance junction wherein several independent resonances can intersect on the constant energy surface. Depending on the coupling regime, the junctions can give rise to local pockets of chaos. Classical trajectories can also become trapped for a finite amount of time leading to several interesting and dynamically relevant consequences [62,63,64,65,66,67,68]. Similarly, concepts such as trapping due to partial barriers and “sticky” dynamics in f≥3 have been investigated [62,69,70,71,72,73] over the past decade in some detail. The jury is still out, but the indications are that there are substantial mechanistic differences in the transport mechanism for f≥3 when compared to the fairly well-understood f=2 case.

The key question, then, is to what extent are the proposed low-dimensional mechanisms for RAT and CAT valid for f≥3 systems? Does this connectedness of the phase space lead to novel mechanisms for DT? That there must be some nontrivial consequences for DT is evident from a very early, and possibly the first, study [49] on a model f=3 system. There, it was explicitly shown that quantum state mixing due to RAT can be clearly understood from the structures on the Arnold web. More recently, it has been shown [52] using a model 4D map that for f≥3 one should anticipate the tunneling enhancements to show complicated peak structure due to the presence of resonance junctions (double or rank-2 resonance) and even drastic suppression of tunneling. The importance of resonance junctions to DT has also been brought up recently in model Hamiltonians relevant for IVR [51] and trapped ultracold atoms [19]. Thus, although there is some progress, answering the questions posed above in general presents several challenges. First, visualization of the phase space is nontrivial but necessary to some extent in order to ascertain the local structures present near the location of the initial quantum state of interest. There has been some progress recently in this regard [74,75]. Second, constructing the Arnold web at a level of detail concomitant with the effective Planck constant *ℏ* (see next section) is a numerically demanding task. Thirdly, given that the resonances are dense everywhere on the web, an estimate of the classical transport timescale connecting two or more quantum states that are involved in DT is needed. This is important to establish if the state mixing is purely quantum mechanical or not. Substantial progress [53,60,76,77,78] has been made recently in terms of estimating the timescale for various model Hamiltonians in the context of Arnold diffusion and Nekhoroshev stability. However, attempts to adapt such techniques to models wherein DT can occur are still lacking.

In the context of the remarks made above, several studies have focused on searching for explicit quantum signatures of the novel phase space transport mechanisms. For instance, Martens analyzed [79] a model three-resonance Hamiltonian to see if the excited eigenstates are delocalized along the resonance channels. Although such delocalized eigenstates were observed, whether or not one can associate them with the Arnold diffusion alone was not clear. In fact, a recent study [51] on the same system (see next section) shows that the observed delocalization can also be due to extensive dynamical tunneling. Leitner and Wolynes quantized the three-resonance model (also known as the stochastic pump model) and noted [80] the equivalence to transport along a disordered wire. Consequently, for any finite value of *ℏ*, quantum localization was predicted. Importantly, localization length was shown to scale as $\hbar^{-3}$, and arguments were provided for the possibility of enhanced transport near the intersection of two independent resonances on the Arnold web. Manifestation of Arnold diffusion in quantum systems has also been studied by Malyshev and coworkers in a series of papers [81,82,83]. It was concluded that if the density of states inside the chaotic layers is large enough (so-called Shuryak border) then quantum Arnold diffusion can occur. Note that an example [84] in the molecular context also indicates that quantum selection rules may limit the extent of diffusion. However, it has also been pointed out [82] that this threshold may not be crucial in driven systems. Indeed, extended diffusion has been observed in a driven two-dimensional optical lattice [85] model. Nevertheless, as concluded by Leitner and Wolynes earlier [80], “quantum” Arnold diffusion is weaker than the classical counterpart due to quantum localization effects. The fact that a combination of quantum localization and novel classical transport can have profound effects has been brought out very nicely by the Dresden group. For example, Stöber et al., in their recent study [86] on coupled kicked rotors, have shown that partial barriers in 4D maps are more restrictive for quantum transport when compared to the 2D maps. A further example comes from the work of Schmidt et al. wherein, using a “synthetic” Hamiltonian, it has been argued [87] that classical drift along a sufficiently wide resonance channel can destroy quantum localization. Consequently, quantum dynamics ensuing from an initial quantum state can explore large regions of the Arnold web. Please note that such extensively delocalized eigenstates have been observed earlier [51] in the context of the Martens model. Moreover, studies [88,89] do indicate that nonlinear interactions can destroy quantum localization.

In what follows, a few of the models are presented along with the key observations. The discussions are by no means exhaustive and certainly no replacement for the original literature, but they do highlight the complexity of DT in f≥3 cases. The review ends with a partial list of questions that remain unanswered.

## 2. Arnold Web: Definition, Construction, and Examples

Given the importance of the Arnold web to DT in f≥3 systems, it is imperative to start with a definition of the web and the generic features. For this purpose, consider a general Hamiltonian of the form
(1)$$H\left(J,\theta \right)=H_{0}\left(J\right)+\epsilon V\left(J,\theta \right)$$
with $\left(J,\theta \right)\equiv \left(J_{1},J_{2},\ldots ,J_{f},\theta_{1},\theta_{2},\ldots ,\theta_{f}\right)$ being the action-angle variables of the *f*-degrees of freedom system. The zeroth-order part $H_{0}$ is assumed to be non-degenerate and integrable. The perturbation is denoted by V(J,θ) with ϵ representing the strength of the perturbation. Typically, for ϵ≠0, the system is nonintegrable, and depending on the perturbation strength, the phase space may vary from being near-integrable to strongly chaotic. Please note that in many instances, one may not be able to explicitly determine the canonical transformations that bring the Hamiltonian to the above form. Nevertheless, for near-integrable systems and in the context of RAT, the Hamiltonian in Equation (Equation 1) is an appropriate starting point. As we see below, the classical limit Hamiltonians corresponding to the Bose–Hubbard model for trapped cold atoms and effective spectroscopic model for molecules are naturally of the form considered. Moreover, the correspondence $J_{k}↔\left(n_{k}+\mu_{k}/2\right)\hbar$ between the classical actions and the quantum numbers $n_{k}$, with the associated Maslov index $\mu_{k}$, provides a convenient platform to compare and contrast the classical and quantum dynamics. From a zeroth-order perspective, one can then define the nonlinear frequencies
(2)$$\Omega_{k}^{0}\left(J\right)\equiv \frac{\partial H_{0}\left(J\right)}{\partial J_{k}}$$
which depend on the actions due to the condition of non-degeneracy. The various frequencies can satisfy commensurability conditions of the form
(3)$$r^{\left(\alpha \right)}\cdot \Omega^{0}\left(J\right)=\sum_{k=1}^{f}r_{k}^{\left(\alpha \right)}\Omega_{k}\left(J\right)\approx 0$$
with $r^{\left(\alpha \right)}=\left(r_{1}^{\left(\alpha \right)},r_{2}^{\left(\alpha \right)},\ldots ,r_{f}^{\left(\alpha \right)}\right)$ being an integer vector. The condition in Equation (Equation 3) represents a nonlinear resonance in the action space of order $O_{\alpha }=\sum_{k=1}^{f}\left|r_{k}^{\left(\alpha \right)}\right|$ with a width scaling as $\sqrt{\epsilon }$ and exponentially with the order. Typically, low-order resonances dominate the early time dynamics, whereas high-order resonances become important for longer periods. In the quantum context, one must also compare the effective *ℏ* with the resonance width to assess the importance of the specific resonance to the dynamics. Interestingly, this dynamical hierarchy of the resonances in terms of their order plays a crucial role in modeling the IVR dynamics in large polyatomic molecules [90].

The resonances defined by Equation (Equation 3) are hypersurfaces in the action space that can intersect the constant zeroth-order energy surface $H_{0}\left(J\right)=E$. For f=2 the intersections are at isolated points, whereas for f≥3, the resonances are no longer isolated, and as seen in Figure 1a, giving rise to a connected network of resonances known as the Arnold web. In Figure 1b, an enlarged portion of the web is shown as an example to indicate that the resonances are dense everywhere and form several resonance junctions. Thus, the resonances of various orders form an intricately connected region over which classical and quantum transport can occur. This aspect implies that any initial state is bound to be under the influence of several resonances. Nevertheless, as indicated in the introduction, one anticipates that only resonances up to a certain maximal order might be relevant for the timescale of interest. What determines this maximal order? Nekhoroshev’s theory [91] is ideally suited for answering this question. In this approach, one restricts attention to resonances up to a maximum order $O_{\alpha }=K\left(\epsilon \right)$. Thus, as sketched in Figure 2 for f=3, the Arnold web can now be divided into various domains. The no-resonance domain comprises all points in the action space that are sufficiently far from resonances of order *K*. In this case, the Hamiltonian is integrable, and frequencies do not vary with time except for an exponentially small diffusion caused by resonances of order higher than *K*. In the single resonance domain, it is possible to transform the Hamiltonian to an integrable single resonance of order $O_{\alpha }\leq K$. One has a fast bounded drift transverse to the resonance line. In the double resonance domain, one has two independent resonances intersecting to form a junction, and the resulting system is nonintegrable. The chaotic motion is bounded and can happen in the region around the junction. Please note that for f>3 one has resonance planes, and it is possible to have m≤f−1 independent resonances that can intersect to form rank-*m* (or multiplicity-*m*) junctions.

The above picture of working with a finite set of resonances leads to Nekhoroshev’s famous stability estimate. For ϵ≪1, an initial condition (J(0),θ(0)) on the Arnold web satisfies
(4)$$\left|J\left(t\right)-J\left(0\right)\right|\leq J_{0}\epsilon^{a}$$
for $0<\epsilon <\epsilon_{0}$ and times
(5)$$t\leq t_{0}\exp\left[{\left(\frac{\epsilon_{0}}{\epsilon }\right)}^{b}\right]$$
with $\left(J_{0},t_{0}\right)$ being positive constants. The stability exponents are estimated to be a=b=1/2(f−m) in the resonance domains. The bound in Equation (Equation 4), apart from indicating stability on an exponential time scale, also implies increased stability near resonance junctions. Interestingly, the stability increases with the increasing rank of the junction. However, in the present context, the relevant timescale is that of DT, which is determined by the precise set of resonances that mediate RAT. Thus, given the sensitivity of RAT to even fairly weak and high-order resonances, an a priori knowledge of the maximal order is not obvious. In addition, for systems with small effective *ℏ*, the density of near-degenerate states is high near junctions with possible involvement of CAT due to the bounded chaos in the vicinity of the junctions. In fact, provided CAT is occurring, the DT timescale may be considerably short, and hence, interesting competition between classical and quantum transport may manifest near the junctions.

The zeroth-order picture is valid for ϵ≪1, and with increasing perturbation strength, the resonances widen, leading to overlaps and the generation of large regions of chaos. The system transitions from the Nekhoroshev regime to the Chirikov regime, as shown in Figure 3 for the example of a model Hamiltonian [94]. In the deep Chirikov regime, there is perhaps no meaningful way to define DT, a statement that is true even in the f=2 case. Theoretically, the former regime, flanked by the Kolmogorov-Arnold-Moser (KAM) and Chirikov regimes, is fairly narrow, and one may rightfully question if a typical physical system can be in such a regime. However, as noted [95] by Morbidelli and Froeschlé, in practice, there is a wider range of ϵ value which characterizes the Nekhoroshev regime. Consequently, studies on DT in model systems in the vicinity of rank-*m* junctions are relevant.

### Construction of the Arnold Web

There are several methods to numerically construct the Arnold web. The essence is to use a measure that can unambiguously distinguish between non-resonant KAM tori, resonance zones, and chaotic regions. Although computing the Lyapunov exponents would be ideal, the numerical overhead is rather large. Consequently, there is considerable interest in numerical approaches that are relatively fast and modest in efforts to map out vast regions of the Arnold web quickly. This would then allow for further studies on the classical transport timescales and comparison with quantum dynamics of specific initial states.

The example shown in Figure 3 is constructed using the method of fast Lyapunov indicator (FLI). The advantage of using the FLI is that one can use finite time dynamics to distinguish between the different dynamical regions. A review of the FLI approach can be found in the original literature. The FLI belongs to a class of variational methods and other measures such as the orthogonal [96] FLI (OFLI), mean exponential growth of nearby orbits [97] (MEGNO), small/general alignment index [98] (SALI/GALI), and relative Lyapunov indicator [99] (RLI) has been proposed. We refer the reader to the review [100] for a comparison of the different chaos indicators and a recent compendium of articles [101] for further information. More recently, Giordano and Cincotta have introduced [102] the Shannon entropy as an efficient measure to construct the Arnold web. Other approaches like the maximum eccentricity-based method [103], frequency map analysis [56,62,104] and wavelet-based measures [105,106,107] provide a fairly powerful approach for construction of the Arnold web.

It is worth noting that, to date, the focus has been on mapping the Arnold web for f=3 Hamiltonian systems. In this case, one can still project the web on an appropriate two-dimensional space, for example, the two independent frequency ratio space [62]. However, for f>3 the situation becomes more complicated, and a different approach, such as the one proposed [108] by Fuji and Toda, may prove useful. In addition, anticipating the increased numerical effort, techniques such as the one based on using the graphic processing unit [109] and Lyapunov weighted dynamics [110] might be more appropriate.

## 3. Dynamical Tunneling and the Arnold Web: Some Examples

### 3.1. Martens’ Three-Resonance Model

The model Hamiltonian introduced by Martens [79] is a fairly good one to study various aspects of DT. The quantum Hamiltonian is given by
(6)$$H=\sum_{i=1}^{3}\left[\omega_{i}\left(a_{i}^{†}a_{i}+\frac{1}{2}\right)+\frac{1}{2}\alpha_{i}{\left(a_{i}^{†}a_{i}+\frac{1}{2}\right)}^{2}\right]+\sum_{i=1}^{3}\tau_{i}V_{i}$$
where the perturbation terms are given by
(7)$$\begin{array}{ccc} V_{1} & = & {\left(a_{1}^{†}\right)}^{2}a_{2}+a_{1}^{2}a_{2}^{†}\\ V_{2} & = & {\left(a_{1}^{†}\right)}^{3}a_{2}^{2}+a_{1}^{3}{\left(a_{2}^{†}\right)}^{2}\\ V_{3} & = & a_{2}^{†}a_{3}^{2}+a_{2}{\left(a_{3}^{†}\right)}^{2} \end{array}$$

The operators $a_{i},a_{i}^{†}$, and $a_{i}^{†}a_{i}$ are the destruction, creation, and number operators. One can imagine the model in Equation (Equation 6) to be an effective “rotating-wave” limit approximation to a more general Hamiltonian. The mode frequencies and anharmonicities are denoted by $\omega_{i}$ and $\alpha_{i}$, respectively. The zeroth-order quantum states are the Fock states $|n_{1},n_{2},n_{3}\rangle$ with the associated zeroth-order energies $E_{n}^{0}$.

The dynamics of the various initial Fock states can then be studied for a wide range of coupling strengths using measures such as the inverse participation ratio and survival probability. Such a detailed study is described in the recent review [7]. However, the focus here is on a specific class of initial states that are involved in DT. Therefore, it is important to study classical dynamics since otherwise, it is not possible to unambiguously associate DT with quantum dynamics. For this purpose, the classical limit of the Hamiltonian Equation (Equation 6) is constructed using the Heisenberg correspondence
(8)$$a_{k}↔\sqrt{J_{k}}\exp\left(-i\theta_{k}\right)\;;\;a_{k}^{†}↔\sqrt{J_{k}}\exp\left(i\theta_{k}\right)$$

The classical Hamiltonian can be expressed as
(9)$$H\left(J,\theta \right)=\sum_{i=1}^{3}\left[\omega_{i}J_{i}+\frac{1}{2}\alpha_{i}J_{i}^{2}\right]+2\sum_{\alpha =1}^{3}\tau_{\alpha }g_{\alpha }\left(J\right)\cos\left(r^{\left(\alpha \right)}\cdot \theta \right)$$

It is easy to check that the above Hamiltonian is a f=3 system since there are no conserved quantities except the total energy. Using Equation (Equation 2), the three different resonance planes and their intersection with the constant energy surface $H_{0}\left(J\right)\approx E$ yields the zeroth-order Arnold web. For concreteness, at this stage we choose the zeroth-order Hamiltonian parameters to be $\left(\omega_{1},\omega_{2},\omega_{3}\right)=\left(1.1,1.7,0.9\right)$ and $\left(\alpha_{1},\alpha_{2},\alpha_{3}\right)=\left(-0.0125,-0.02,-0.0085\right)$ in scaled units. The parameters are essentially chosen so that various structures on the Arnold web at the energy of interest can manifest. Thus, by varying the parameters of the zeroth-order Hamiltonian, one can “engineer” different scenarios in terms of the location of the single resonances and the total number of resonance junctions. An example is shown in Figure 4 wherein the different web structures with changing $\omega_{2}$ and *E* can be clearly seen. For example, at E=20 and $\omega_{2}=1.3$, the resonance planes do not manifest, and hence there is no web structure expected. On the other hand, for $\omega_{2}=1.5$ and E=40, all three resonances can be seen, and one of the junctions appears around $\left(J_{1},J_{2}\right)\approx \left(28,0\right)$, i.e., at the “edge” of the action space. Similarly, at $\omega_{2}=1.9$ the resonances $R_{3}$ and $R_{1}$ intersect for E=30, whereas they do not intersect for E=20. We mention, without going into details, that there are certain conditions known as steepness in order for the Nekhoroshev theorem to hold. The Martens’ model does not satify the steepness condition. However, suffice it to note that for the parameter choices made, our system is quasi-convex in the action region of interest.

The zeroth-order analysis and expectation can be made more precise by numerically mapping the Arnold web using the FLI technique. Details of the computation can be found in the earlier publication [51]. Briefly, a large grid of initial conditions on the $\left(J_{1},J_{2}\right)$ plane is selected for a specific angle slice. The action $J_{3}$ is selected using energy conservation, and the resulting ensemble of initial conditions is propagated to a sufficiently large time so that the FLI can clearly distinguish between the different dynamical behaviors. As an example, the Arnold web for a total energy E≈40 is shown in Figure 5, indicating the existence of two prominent resonance junctions labeled A and B. These arise from a crossing of the $r^{\left(3\right)}\equiv \left(0,1,-2\right)$ (denoted as $R_{3}$ in Figure 5) with the $r^{\left(1\right)}\equiv \left(2,-1,0\right)$ (denoted as $R_{1}$) and $r^{\left(2\right)}\equiv \left(3,-2,0\right)$ (denoted as $R_{2}$) resonances, as also predicted by the zeroth-order analysis in Figure 4. Note that the web is sparse because we have picked a model with exactly three primary resonances Alternatively, one can think of the Hamiltonian arising from restricting the resonancesto a maximum order, as in the Nekhoroshev approach.

A few points are worth noting. First, Figure 5 is not close to the Chirikov limit yet. Nevertheless, the chaotic regions near the rank-2 resonances are evident (see the inset). Second, the two junctions are well separated, which is ideal for investigating RAT in such regions. Third, previous work [7] has shown that the dynamics near the two junctions are quite different. With the Arnold web structure established for the given energy and coupling values, we can now study the RAT mechanism far from and near a specific resonance junction. In Figure 6, we show the quantum and classical dynamics of the initial Fock state |n〉=|22,1,19〉 in terms of the survival probability
(10)$$P_{n}\left(t\right)\equiv {\left|\langle n|\exp\left(-iHt/\hbar \right)|n\rangle \right|}^{2}$$

The location of the initial state on the Arnold web is also shown in the figure. Clearly, the initial state is in the vicinity of the $R_{1}$ resonance and away from the junction. Figure 6 shows that this is DT mediated by the nonlinear resonance since the classical dynamics is localized. This is also clear from the classical dynamics projected onto the Arnold web (green dot). In contrast to the classical dynamics, the quantum counterpart shows coherent oscillations with a period of about T∼1300 (∼400T_2_ in terms of the harmonic mode time period $T_{2}=2\pi /\omega_{2}$) and nearly mimics a two-state Rabi oscillation. Further analysis shows that the second state involved in the quantum dynamics is |20,2,19〉 which lies nearly symmetric about the $R_{1}$ resonance center line (indicated by a red dot on the Arnold web in Figure 6). Further analysis shows that the observed DT can be accounted for using the RAT theory [35] involving the $R_{1}$ resonance.

How does the above single resonance picture change if the initial Fock state is located close to the $R_{1}-R_{3}$ resonance junction? Given the fact that an infinity of resonances of various orders exists at the junction, one anticipates a more complicated picture when compared to the above single resonance case. This is illustrated in Figure 7 for the dynamics of the initial state |25,4,9〉. For low coupling strengths Figure 7a shows that there are three other states $|s_{1}\rangle =|23,5,9\rangle$, $|s_{2}\rangle =|25,3,11\rangle$, and $|s_{3}\rangle =|23,3,13\rangle$ that mix significantly. These states do not mix classically, and hence, quantum mixing is an example of DT. The mixing between the initial state and the states $|s_{1}\rangle$ and $|s_{2}\rangle$ can be associated with RAT mediated by the resonances $R_{1}$ and $R_{3}$, respectively. However, the state $|s_{3}\rangle$ mixes coherently on a timescale of ∼10,000 T_2_. This is a clear influence of the junction since one can show that the (2,0,−2) induced resonance at the junction mixes $|s_{2}\rangle$ and $|s_{3}\rangle$. This induced resonance is visible in Figure 7a, and the timescale is much longer due to the effective coupling strength being $\tau_{1}$$\tau_{3}$∼10^−9^, i.e., nearly four orders of magnitude smaller than the primary resonances. Despite this, it is observed that the populations of all three states involved are nearly the same (∼15 %) at ∼10,000 T_2_.

A more surprising and key aspect of the influence of a junction on DT occurs upon increasing the resonance strengths. As shown in Figure 7b, increasing the $R_{1}$ and $R_{3}$ resonance strengths leads to many more states that mix due to DT. However, some of the states mix classically as well. In fact, states $|s_{1}\rangle$ and $|s_{2}\rangle$ are now classically connected on timescales similar to the quantum. However, the quantum probabilities are larger and inverted relative to the classical result. Moreover, new states like $|s_{4}\rangle =|21,5,11\rangle$ gain significant populations (∼30%) within a timescale of about ∼800 T_2_ whereas the state $|s_{3}\rangle$, although still mixing solely due to DT, only gains about ∼5%. Note that the suppression in the population of $|s_{3}\rangle$ happens despite the effective resonance strength being nearly an order of magnitude larger than in Figure 7a. Perhaps this suppression comes about due to the “canceling paths” proposed in the recent work [52] of Firmbach et al. Confirming this requires further study in terms of an appropriate effective Hamiltonian near the junction of interest. It is expected that variation of the effective *ℏ* can lead to a better understanding of the results in Figure 7b. However, note that this is numerically challenging since the density of states increases rather rapidly for the Martens’ model. Thus, for $\hbar_{\mathrm{eff}}$∼0.01 one may need to diagonalize very large matrices even for restricting attention to eigenstates in a narrow energy range. For example, with ℏ=1, there are nearly 150 near-degenerate states for Δ E ∼0.1 around E=40. In any case, the model Hamiltonian in Equation (Equation 9) needs further studies over a wider parameter range to bring out the influence of the Arnold web on the DT process.

### 3.2. Trapped Ultracold Atoms

Another system wherein DT is expected to play a significant role is the optically trapped ultracold atoms [111,112] which can be usefully analyzed in terms of the Bose–Hubbard Hamiltonian (BHH).There is an interesting parallel between the BHH and the effective spectroscopic Hamiltonian of the form in Equation (Equation 6)—The number of sites (wells) in an optical trap and the number of particles on each site correspond with the number of vibrational modes and the excitation quanta of each mode, respectively. Thus, *N* particles trapped in a (f+1) site potential can be described by a *f* degrees of freedom Hamiltonian since the total particle number is conserved. The hopping terms in the BHH correspond to nonlinear resonances in the classical limit, which is approached for many trapped atoms since *ℏ*_eff_∼*N*^−1^. For the 2-site BHH studies have shown that one can predict and experimentally observe [113] interesting phases such as the macroscopic quantum self-trapping (MQST) phase by analyzing the classical limit Hamiltonian [114]. In particular, MQST arises due to the interplay between the hopping (tunneling) and the interaction strengths. Wüster et al. have shown [115] that MQST also emerges in the context of dynamical tunneling of a driven Bose–Einstein condensate in a single well. It is, therefore, interesting to ask if other novel phases can emerge in multi-site BHH models and if the existence of such phases can be correlated with the features on the Arnold web.

Clearly, the first requirement of addressing the question above is to construct the Arnold web, and a minimal model is a 4-site system. Recently [51] such a system was analyzed where the BHH $H=H_{T}+H_{M}$ was considered with
(11)$${\hat{H}}_{T}=\frac{U}{2}\sum_{j=1}^{3}{\hat{n}}_{j}^{2}-\frac{K}{2}\sum_{j=2,3}\left({\hat{a}}_{j}^{†}{\hat{a}}_{1}+h.c.\right)$$
and
(12)$${\hat{H}}_{M}=\frac{U}{2}{\hat{n}}_{0}^{2}-\frac{K_{c}}{2}\sum_{j=1}^{3}\left({\hat{a}}_{j}^{†}{\hat{a}}_{0}+h.c.\right).$$

The model above is taken from the work [116] of Khripkov, Cohen, and Vardi. The site energies are denoted by *U*, and $K,K_{c}$ are the hopping amplitudes. Essentially, as indicated in Figure 8 inset, $H_{T}$ describes a 3-site linear trimer coupled to a monomer via $H_{M}$. Please note that for $K_{c}=0$ the monomer decouples from the system and $X\equiv n_{1}+n_{2}+n_{3}$ is a conserved quantity. On the other hand, for $K_{c}\neq 0$, the conservation of *X* is violated, but the total particle number $N\equiv X+n_{0}$ is conserved. Thus, the eigenstates of the full Hamiltonian can be expressed as a linear combination of the Fock states $\left|n;N\rangle \equiv \right|n_{1},n_{2},n_{3};N\rangle$.

An aspect of interest for such bipartite models is to compare classical versus quantum thermalization [117] triggered by a weak monomer coupling. For instance, Figure 8 shows the extent to which eigenstates of the trimer are delocalized in the *X* direction since [H,X]≠0 for finite values of $K_{c}$. Three example trimer eigenstates are shown, and it is clear that the spreading in the *X* direction can be extensive for certain states. The question is whether this spreading in *X* is entirely due to DT or whether there is some classical contribution as well. To address this issue, one can study the dynamics of specific initial states |n;N〉 for $K_{c}\neq 0$, particularly those that contribute dominantly to the trimer eigenstate spreading seen in Figure 8.

Among the several initial states studied in a recent work [19], we illustrate the dynamics of the state |16,0,8;40〉. This state is chosen since it is representative of the class of states for which the dynamics have both classical and quantum contributions. As seen from Figure 9, for $K_{c}=0$, the state is localized and not affected by the monomer. For finite $K_{c}$, the trimer is perturbed by the monomer, and the quantum survival probability decays, exhibiting multiple timescales. The shortest timescale in Figure 10a, of *Kt*∼0.5 shows coherent oscillations involving the initial state and two other states due to the $a_{1}a_{0}^{†}$ hopping term. The analogous classical computations shown in Figure 10b indicate that there is a flow to the states corresponding to Figure 10a, albeit on a longer timescale.

On the other hand, Figure 10c shows that the longer timescale of *Kt*∼100 seen in Figure 9 for $K_{c}\neq 0$ correlates with significant population flow into multiple number of states. However, as Figure 10d shows, there is no classical probability flow to the states in Figure 10c, even on fairly long timescales. Thus, Figure 10b,c represent classes of states that are connected and not connected by the classical flow, respectively. It is also worth noting that while the quantum dynamics exhibits coherent oscillations over a timescale of *Kt*∼500, the classical dynamics “thermalizes” by *Kt*∼20. Thus, there is a distinct difference between the classical and quantum dynamics of the initial state of interest.

Understanding the results shown in Figure 10 requires a careful study of the classical dynamics. As before, using the Heisenberg correspondence, the classical Hamiltonian can be expressed as
(13)$$H\left(J,\theta \right)=H_{0}\left(J\right)+V\left(J,\theta \right)$$
with $H_{0}\left(J\right)\equiv U\sum_{j=0}^{3}J_{j}^{2}/2$ and
(14)$$V\left(J,\theta \right)\equiv -K\sum_{j=2,3}\sqrt{J_{1}J_{j}}\cos\theta_{1j}-K_{c}\sum_{j=1}^{3}\sqrt{J_{0}J_{j}}\cos\theta_{0j}$$
where we have denoted $\theta_{kl}\equiv \theta_{k}-\theta_{l}$. Using the zeroth-order nonlinear frequencies Equation (Equation 2), the five primary resonances of the above Hamiltonian can be determined along with their projections on a specific set of action planes of interest. For instance, in $\left(I_{2},I_{3}\right)$ space the three trimer-monomer resonances (denoted $R_{Mk}$) can be expressed as
(15)$$\begin{array}{ccc} \Omega_{0}\left(J\right) & = & \Omega_{1}\left(J\right)\Rightarrow R_{M1}:J_{3}=\left(2X_{c}-N\right)-J_{2}\\ \Omega_{0}\left(J\right) & = & \Omega_{2}\left(J\right)\Rightarrow R_{M2}:J_{2}=N-X_{c}\\ \Omega_{0}\left(J\right) & = & \Omega_{3}\left(J\right)\Rightarrow R_{M3}:J_{3}=N-X_{c} \end{array}$$
whereas the two resonances within the trimer subspace are
(16)$$\begin{array}{ccc} \Omega_{1}\left(J\right) & = & \Omega_{2}\left(J\right)\Rightarrow R_{T1}:J_{3}=X_{c}-2J_{2}\\ \Omega_{1}\left(J\right) & = & \Omega_{3}\left(J\right)\Rightarrow R_{T2}:2J_{3}=X_{c}-J_{2} \end{array}$$

In the above $X_{c}\equiv \sum_{k=1}^{3}J_{k}$ and $N\equiv J_{0}+X_{c}$ being the classical analog of the quantum total particle number *N*. The expectation is that if the initial state is in the vicinity of the junctions formed by the intersection of $R_{Mk}$ and $R_{Tk}$, then substantial perturbation of the trimer dynamics can occur for $K_{c}\neq 0$. Moreover, several RAT pathways can open up at the junction and result in the multiple timescales seen in Figure 9.

To confirm the above “suspicion”, we construct the Arnold web for specific $\left(X_{c},N\right)$ using the FLI technique. Computations show that varying the trimer population $X_{c}$ for the fixed total number of particles reveals the Arnold web [19] structure changing in terms of the type and number of resonance junctions. A typical web with several junctions is shown in Figure 11a for X=36 and N=40. In the context of the dynamics shown in Figure 9 and Figure 10, the relevant portion of the Arnold web in $\left(J_{1},J_{0}\right)$ space is shown in Figure 11b along with the location of the initial state. The quantum dynamics of the initial state are shown in terms of the probability flow through the quantum number space up to a maximum time *Kt*∼1000. A few observations can be made at this stage. First, the initial state is located within the $R_{M1}$ resonance. Consequently, the population can resonantly transfer between the monomer and site 1 of the trimer. Second, significant delocalization can be observed around the resonance junction. However, on this timescale, it is clearly non-uniform—the probability of population transfer to the monomer is larger. Thirdly, the dynamics for the various states shown in Figure 10b,c are clearly identified on the Arnold web and confirm the role of the resonance junction. Specifically, the states connected by double arrows in Figure 11b are precisely the ones that are involved in DT in Figure 10c. Thus, based on the observed dynamics near the junction, a possible dominant path that connects the states $|n_{1},n_{2},n_{3};N\rangle =|16,0,8;40\rangle$ with the state |8,0,16;40〉 is as follows 
$$|16,0,8;40\rangle ⟶|15,0,8;40\rangle \overset{\mathrm{DT}}{\to }|13,0,9;40\rangle ⟶|9,0,13;40\rangle \overset{\mathrm{DT}}{\to }|8,0,16;40\rangle$$where paths are occurring due to DT, and hence classically forbidden, are indicated. The equivalent chain of paths in the $|n_{0},n_{1},n_{2};N\rangle$ representation is 
$$|16,16,0;40\rangle ⟶|17,15,0;40\rangle \overset{\mathrm{DT}}{\to }|18,13,0;40\rangle ⟶|18,9,0;40\rangle \overset{\mathrm{DT}}{\to }|16,8,0;40\rangle$$which can be directly correlated with Figure 11b. It can be shown (argued) that the first (last) of the DT paths occur due to RAT involving a third (sixth) order resonance induced at the resonance junction. At this juncture, it is useful to recollect the previous discussion on the issue with a maximal order choice within the Nekhoroshev approach. Clearly, the DT timescales are sensitive to fairly high-order resonances). Hence, the probability flow in Figure 10c is a clear f=3 effect. One may argue that the initial and final states can be connected simply by the $R_{T2}$ resonance. However, this is not very probable since the process involves eight particle exchanges between sites one and three of the trimer. Moreover, if that were to be the case, then the survival probability in Figure 9 would have decayed even for $K_{c}=0$.

Again, the above example is a hint at the possible effect of the resonance junctions on DT. Much more can be learned from this model by looking at wider parameter regimes. A start has been made in the recent work [19], and it would be interesting to study aspects of thermalization in such systems [118] due to the presence of the resonance junctions. Please note that in the context of unimolecular decay reactions, there is [68] already a strong connection between the junctions and non-statistical dynamics.

## 4. Final Thoughts

This review has attempted to highlight the complexity of studying DT in systems with three or more degrees of freedom. Although a fair amount of progress has happened over the past decade, there are still several questions that remain unanswered. Here is a partial list of questions:Almost all the examples shown here suffer from one key issue. There is simply no accurate estimate of classical stability times and their comparison to the DT timescales. Moreover, a careful study of the DT process by scaling the effective *ℏ* needs to be done. In this regard, it may be worthwhile to study Martens’ model from the stochastic pumping (or three-resonance) model perspective.For mixed regular–chaotic phase spaces in f=2, a combination of RAT and CAT is operative. Models combining the nonlinear resonances and random matrix theory have been relatively successful in understanding tunneling splittings. For f≥3, the local chaos near the junctions may not be amenable to a random matrix approach. How does one account for the role of CAT, if relevant, near junctions?The focus, understandably so, has been on f=3 systems. What about f>3 systems? Higher rank junctions are now possible. Moreover, the argument [80] that quantum Arnold diffusion may delocalized in analogy with the transport along disordered wires is no longer valid. Similarly, whether the destruction of quantum localization on the Arnold web due to classical drift [87] holds in the presence of higher rank junctions is not clear at the present moment. Already for f=3, the results in Figure 7 and Figure 10b seem to suggest a stronger Nekhoroshev stability for the quantum dynamics. Of course, one needs to ask: is there a “quantum” Nekhoroshev theorem? Some subtle issues in this regard have been outlined in the paper by Fontanari et al. [119].Much of the arguments invoking the Nekhoroshev exponential stability need modification when the quasi-convexity or steepness assumptions are violated. In such instances, one can have fast transport on the Arnold web. Does this then invalidate the notion of DT in such systems? Even for such systems, are there phase space regions that are classically disconnected over physically relevant timescales? In an impressive study, Pittman, Tannenbaum, and Heller have [50] made a start in terms of non-convex model Hamiltonians. In fact, and relevant to the previous point, they studied DT in systems with f=3,4, and 5 and argued that DT can be faster than the fast classical transport and hint at mechanisms different from RAT. However, certain coupling schemes can result in comparable timescales for classical and quantum transport. More extensive studies on this and other such models would yield important insights.

The list (admittedly partial) of questions above indicates that our understanding of DT in f≥3 systems is still in its infancy. However, answers to the questions are expected to shed light on issues ranging from IVR in polyatomic molecules to thermalization in interacting many body systems.

## Figures and Tables

**Figure 1 entropy-26-00333-f001:**
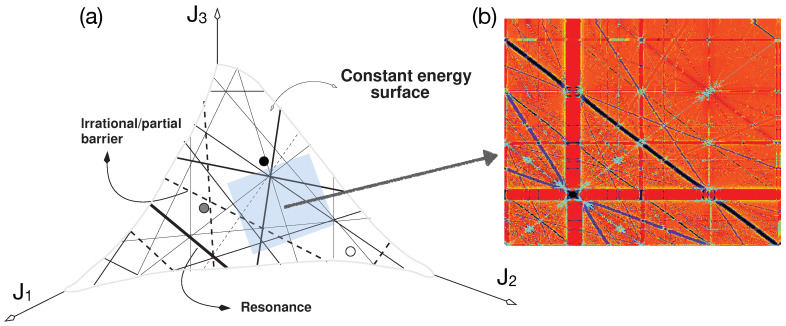
(**a**) A sketch of the Arnold web for f=3 formed by resonance surfaces intersecting the constant energy surface. The circles indicate possible initial classical or quantum states. Partial barriers formed by pairwise noble tori may also be present. (**b**) Enlarging the blue shaded region in (**a**) indicates that the resonances are dense everywhere on the Arnold web.

**Figure 2 entropy-26-00333-f002:**
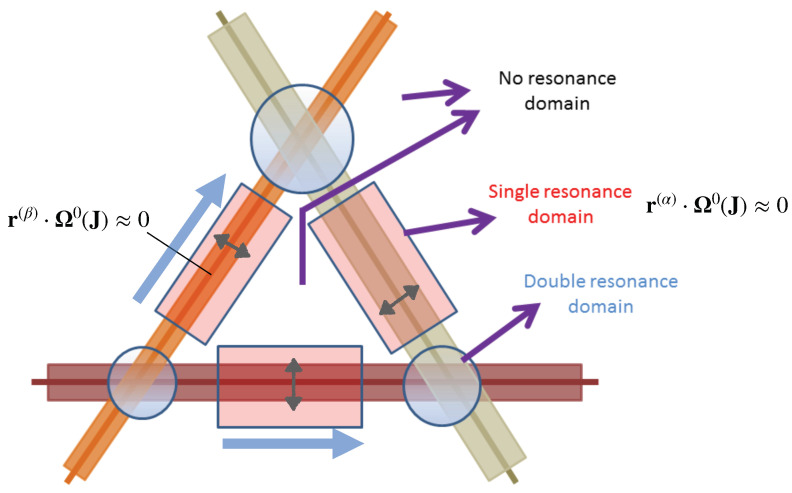
Inclusion of resonances to a certain maximal order. Defining the single, double, and no-resonance domains in Nekhoroshev theory. Fast drift (gray double arrows) occurs transversely to the individual resonances. Exponentially slow Arnold diffusion (thick blue arrow) can occur along the resonance. Figure adapted with permission from the PhD thesis [92] of S. Karmakar, which is based on the figure in [93].

**Figure 3 entropy-26-00333-f003:**
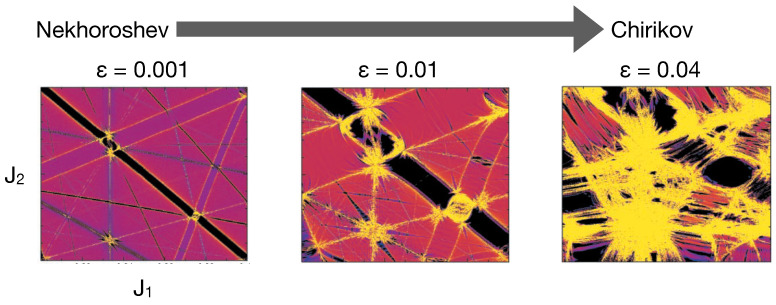
An illustration of the transition from Nekhoroshev to the Chirikov regime with increasing coupling strength ϵ. The figure is adapted from the work [94] of Guzzo et al. wherein the model f=3 Hamiltonian with $H_{0}\left(J\right)=J_{3}+\left(J_{1}^{2}+J_{2}^{2}\right)/2$ and $V\left(J,\theta \right)=\epsilon {\left(4+\sum_{k=1}^{3}\cos\theta_{k}\right)}^{-1}$ is used. The web is constructed using the FLI method. Low FLI values (black) represent the resonance zones. High FLI values in yellow correspond to chaotic motion at the intersection of the resonances or separatrices associated with the single resonance zones.

**Figure 4 entropy-26-00333-f004:**
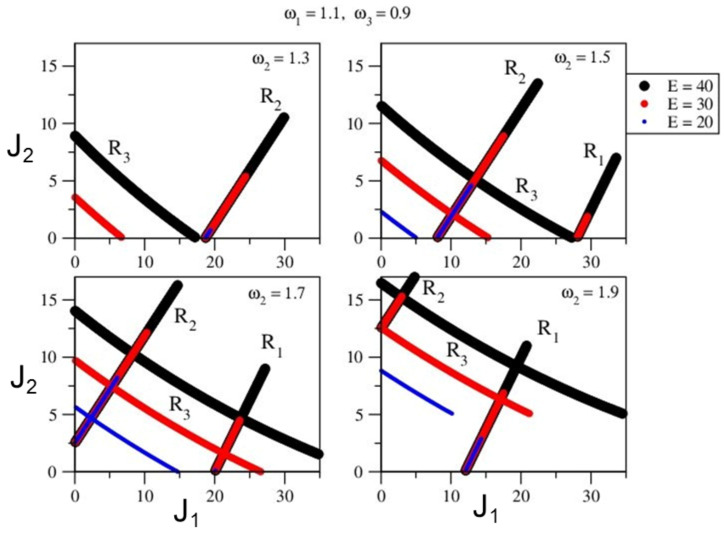
Zeroth-order Arnold web prediction for the model Hamiltonian Equation (Equation 9) with varying total energy *E* and harmonic frequency $\omega_{2}$ of the second mode. The other two mode frequencies are fixed at $\omega_{1}=1.1$ and $\omega_{3}=0.9$. Please note that for every choice of $\omega_{2}$ (a given panel), the resonances at three energies E=20 (small blue circle), E=30 (medium red circle), and E=40 (large black circle) are shown. If a particular color line is missing, it implies that the corresponding resonance does not appear at that energy. Figure taken with permission from the PhD thesis [92] of S. Karmakar.

**Figure 5 entropy-26-00333-f005:**
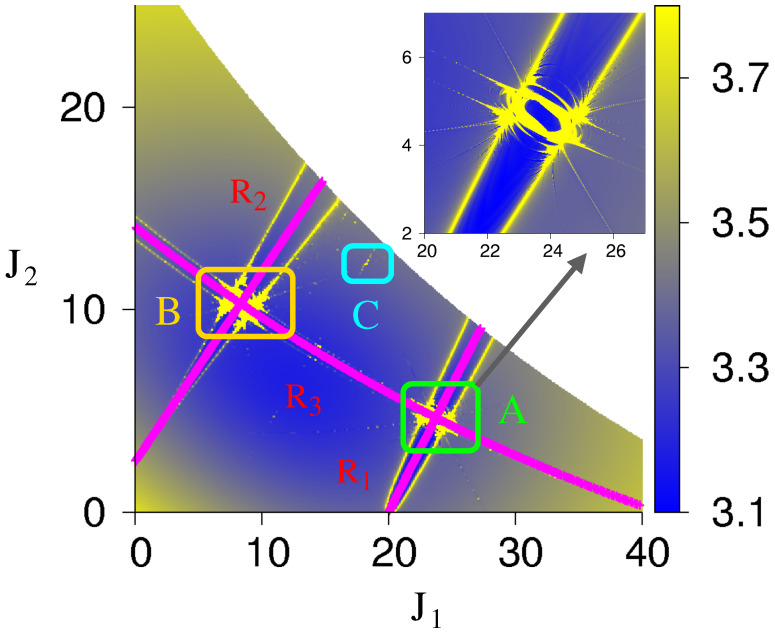
Arnold web for the model Hamiltonian Equation (Equation 9) at total energy E≈40 constructed using the FLI technique. The initial angle slice is (π/2,π/2,π/2) and the resonant coupling strengths are taken as $\left[\tau_{1},\tau_{2},\tau_{3}\right]=\left[5,1,5\right]\times 10^{-5}$. The FLI scale is shown with FLI values greater than 3.7, indicating chaotic regions (in yellow), while the lowest FLI value (in blue) highlights the resonance zone. Two prominent junctions labeled A and B can be seen. The zeroth-order prediction of the resonance center lines is indicated in purple. Another junction, C, arises out of the intersection of higher order and induces resonances. (Inset) An enlarged plot of the region near junction A is shown. The FLI scale is the same as in the main plot. Figure adapted from [51].

**Figure 6 entropy-26-00333-f006:**
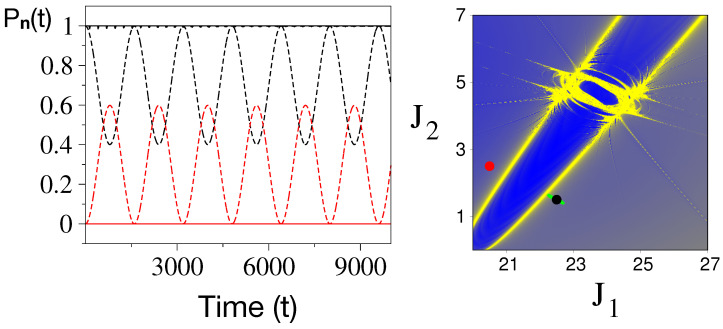
Classical (solid line) and quantum (dashed line) survival probabilities of the initial state |22,1,19〉 (black) and the state |20,2,19〉 (red). The location of the two states on the Arnold web is shown in the right panel. Parameters as in Figure 5 and the projected classical flow of the initial state are shown in green. Please note that the zeroth-order energies of the two states are $E_{22,1,19}^{0}\approx 40.28$ and $E_{20,2,19}^{0}\approx 40.27$. Figure adapted from [51].

**Figure 7 entropy-26-00333-f007:**
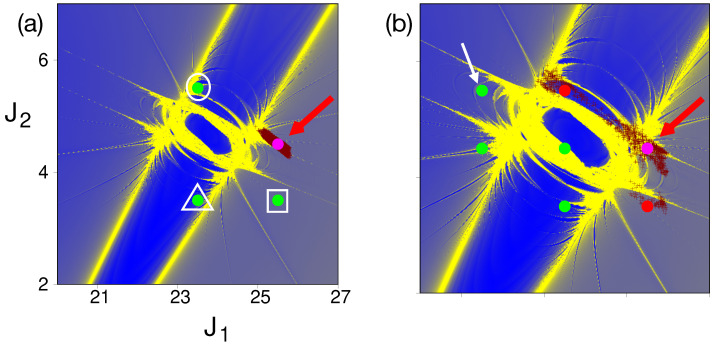
Quantum and classical dynamics for the initial state |25,4,9〉 (shown by red arrow) located near the junction. (**a**) For coupling strengths $\left[\tau_{1},\tau_{2},\tau_{3}\right]=\left[5,1,5\right]\times 10^{-5}$ three other quantum states (shown in green) $|s_{1}\rangle$ (circle), $|s_{2}\rangle$ (square), and $|s_{3}\rangle$ (triangle) participate in the dynamics. The classical dynamics is localized (brown dots) (**b**) For coupling strengths $\left[\tau_{1},\tau_{2},\tau_{3}\right]=\left[10,1,10\right]\times 10^{-5}$ more states participate in the dynamics. However, certain states (in red) mix classically, whereas certain other states (green) mix only quantum mechanically. An example of the latter is state $|s_{4}\rangle =|21,5,11\rangle$ whose location is indicated by a white arrow. The brown dots show a typical classical trajectory projected on the Arnold web. Figure adapted from [51].

**Figure 8 entropy-26-00333-f008:**
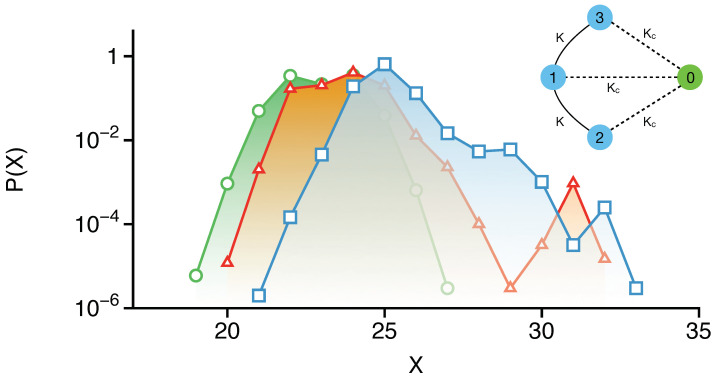
The distribution of $X=n_{1}+n_{2}+n_{3}$ at time Kt=1000 for three example trimer eigenstates upon coupling the monomer. The selected trimer eigenstate belongs to the X=25 (blue squares), X=24 (red triangles), and X=23 (green circles) manifold. (Inset) A schematic of the four-site Bose–Hubbard model with the site numbering used in the text. The value of the parameters used are $U=0.5,K=0.1,K_{c}=0.05$, and N=40. Figure adapted from [19].

**Figure 9 entropy-26-00333-f009:**
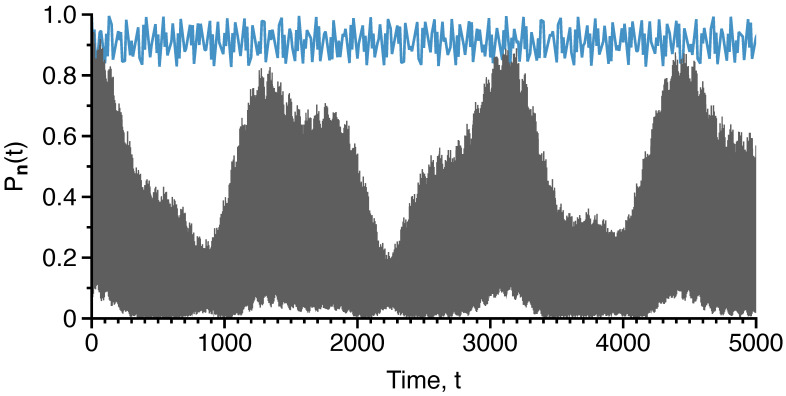
Survival probability of the initial Fock state |16,0,8;40〉 for $K_{c}=0$ (blue) and $K_{c}=0.05$ (gray). Parameters fixed at U=0.5,K=0.1, and N=40. Figure adapted from the PhD thesis [92] of S. Karmakar.

**Figure 10 entropy-26-00333-f010:**
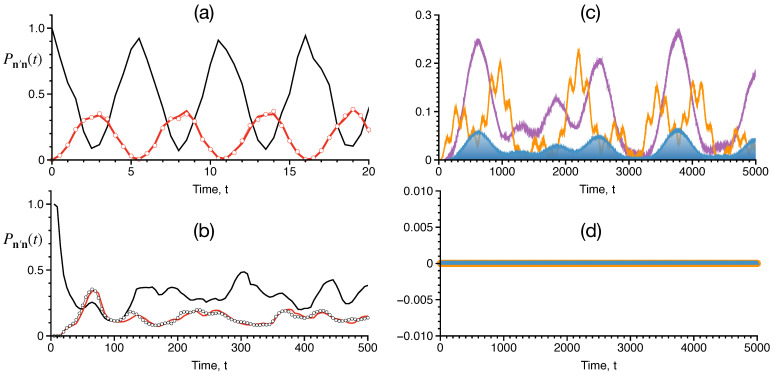
Dynamics of the Fock state |16,0,8;40〉 for parameters U=0.5,K=0.1, and N=40. (**a**) Survival probability of |16,0,8;40〉 (black) and probability flow into states |15,0,8;40〉 (red), |17,0,8;40〉 (circles) on short timescales. (**b**) Classical survival probability of |16,0,8;40〉 (black) and probability flow into states |15,0,8;40〉 (red), |17,0,8;40〉 (circles). (**c**) Quantum probability flow into states |8,0,16;40〉 (purple), |9,0,13;40〉 (orange), and |8,0,15;40〉 (blue shaded). (**d**) Classical analog for the results shown in (**c**) indicates localization. Figure adapted from the Ph.D. thesis [92] of S. Karmakar.

**Figure 11 entropy-26-00333-f011:**
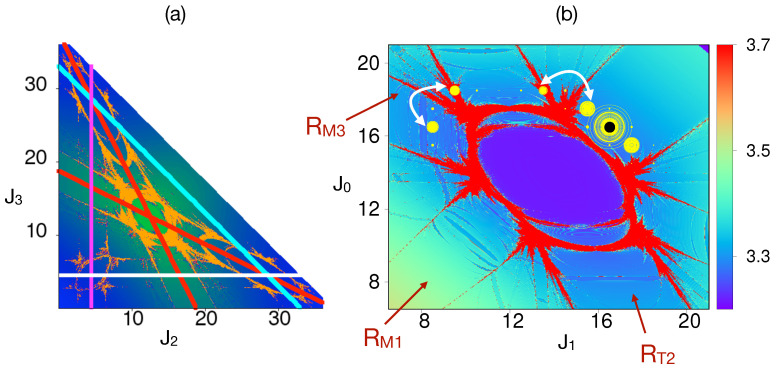
(**a**) An example Arnold web for (X,N)=(36,40) mapped using the FLI technique. The yellow regions represent chaos. Based on the zeroth-order predictions, the trimer subspace resonance $R_{Tk}$ centers are shown in red, and the monomer–trimer resonance $R_{Mk}$ centers are shown in cyan, white, and purple colors. (**b**) A close-up of the resonance junction in $\left(J_{1},J_{0}\right)$ space formed due to the intersection of $R_{M1}$ and $R_{T2}$ resonances. The initial Fock state $|n_{0},n_{1},n_{2};N\rangle =|16,16,0;40\rangle$ (same as the state $|n_{1},n_{2},n_{3};N\rangle =|16,0,8;40\rangle$) is shown as a black dot. The quantum probability flow to different participating Fock states at times Kt=0,10,20,…,1000 are shown as yellow circles with radius ∝ probability. The white arrows connecting a pair of states correspond to classically forbidden but quantum mechanically allowed processes. Figure adapted from [19].

## Data Availability

No new data have been generated for this article, and all the data presented here can be found in the original cited literature.

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
