# Peer review of "Dynamical Tunneling in More than Two Degrees of Freedom"

_entropy, 2024, doi:10.3390/e26040333_

Round 1

Reviewer 1 Report

Comments and Suggestions for Authors

This article provides a comprehensive and rather complete review of theoretical and numerical studies on dynamical tunneling in semiclassical quantum systems that exhibit more than two degrees of freedom. Special emphasis is put on the structure of the Arnold web which displays the location of resonances in the classical phase space. Numerical studies carried out within bosonic model Hamiltonians yield clear signatures for resonance-assisted tunneling within this Arnold web.

The paper is well written and I recommend it for publication in Entropy. I have a few remarks of minor importance, mainly referring to the viewgraphs, that the author might want to take into account:

- In Figure 3 the color code is not explained. I presume the color relates to the FLI as in other viewgraphs. This should be made explicit.

- I have a hard time to understand the panels of Figure 4 where several energy layers seem to be superposed. Perhaps it would be better to split each viewgraphs into subpanels showing the situation for each particular energy.

- In the corresponding caption "... varying total energy $E \approx 40$" does not seem to make sense since in the viewgraphs the energy equals 20 and 30, too.

- Figure 5 is not properly referenced in the text ("Fig. ??").

- In Figure 5 (and subsequent figures) the FLI varies from 3.1 to approximately 3.75. The upper end of this scale is associated with chaos. What about the lower end? Can one argue that FLI=3 equals nearly regular dynamics? Or are the blue regions in the phase-space plot just slightly less chaotic than the resonance junction regions? A comment on this would be helpful for the reader.

- In Figure 7 it would be useful to explicitly indicate the positions of the individual states, even though their coordinates are detailed in the text.

- In Figure 9, the color representation in panel (a) is not optimal. And the inset in panel (d) has axis and tick labels that are unreadable in a printed version of the manuscript.

Author Response

I thank the reviewer for going through the manuscript and providing important comments. Below, I address the points raised by the referee:

  • In Figure 3 the color code is not explained. I presume the color relates to the FLI as in other viewgraphs. This should be made explicit.

  • Thanks. This has been indicated in the caption now.

    - I have a hard time to understand the panels of Figure 4 where several energy layers seem to be superposed. Perhaps it would be better to split each viewgraphs into subpanels showing the situation for each particular energy.

  • I agree with the reviewer. However, a single panel for the three energies was adopted to avoid having 12 separate plots! That was also the reason for the different energies being shown in different colours and point sizes. I have added some additional description in the caption and in the text. Hopefully, this will make it easier to read.

    - In the corresponding caption "... varying total energy $E \approx 40$" does not seem to make sense since in the viewgraphs the energy equals 20 and 30, too.

  • Thanks -- corrected.

    - Figure 5 is not properly referenced in the text ("Fig. ??").

  • Corrected.

    - In Figure 5 (and subsequent figures) the FLI varies from 3.1 to approximately 3.75. The upper end of this scale is associated with chaos. What about the lower end? Can one argue that FLI=3 equals nearly regular dynamics? Or are the blue regions in the phase-space plot just slightly less chaotic than the resonance junction regions? A comment on this would be helpful for the reader.

  • Actually, the upper FLI scale means all values higher than or equal to 3.75. So, one can have even values as large as 10 or 15 represented by the same yellow colour. This has been indicated now. The lower FLI scales can represent regular dynamics of various extent. Distinction between KAM tori and resonant tori are usually finer. But in this figure the object was to exhibit the web clearly showing the various resonance zones and their intersections.

    - In Figure 7 it would be useful to explicitly indicate the positions of the individual states, even though their coordinates are detailed in the text.

  • Done.

    - In Figure 9, the color representation in panel (a) is not optimal. And the inset in panel (d) has axis and tick labels that are unreadable in a printed version of the manuscript.

  • The figure has now been split into two in the revised manuscript. Hopefully, it is easier on the eyes now!

Reviewer 2 Report

Comments and Suggestions for Authors

This manuscript attempts to review the mechanisms for dynamical tunneling in systems with three or more degrees of freedom, where Arnold diffusion dominates the dynamics.

Non-integrable systems with f=3 or more degrees of freedom (f DoF with f>2) are dynamically very different from those with two degrees of freedom.  Systems with f>2 contain a dense web of resonances, called the Arnold web, which permeates the phase space. Most of the work done to understand the dynamics induced by the Arnold web has been done on systems with 3 DoF, because some systems with 3 DoF can be explored numerically. 

This paper does give a fairly complete set of references to work that has been done on systems with three or more DoF, on that basis deserves to be published.  However, there are some issues with the current version that should be corrected before it is published.  These issues are listed below.

(1)     The author distinguishes between “resonance assisted tunneling (RAT)” and “chaos assisted tunneling (CAT)” but never explains the difference.  It would be useful to insert clearer definitions of these two tunneling mechanisms.  It is not clear if CAT actually exists in systems with 3 DoF (or more DoF).  CAT generally involves a symmetry that leads to the quantum tunneling.  Under what conditions do analogous symmetries exist in systems with 3 DoF?

(2)     The author refers to “quantum dynamical localization”.  He should describe in more detail what this is.

(3)     References 2, 51, 52, and 82 appear in footnotes, which puts them out of order and hard to find. Is this acceptable to the journal.

(4)     The author lists reviews of CAT but does include the recent Entropy journal review on CAT:  Entropy 2024, 26, 144.

(5)     The English in the paper needs checking. For example, lines 88 and 173 in the proofs are garbled. 

(6)     The Figure reference on lines 277, 279, 283, and Fig. 6 is faulty.

Comments on the Quality of English Language

The english is fairly good, but some sentences are garbled.  The english should reviewed carefully

Author Response

I thank the reviewer for the comments and suggestions. Below I provide a pint by point response to the reviewer's comments. I would, however, make a remark that none of the three degrees of freedom models discussed in this review have Arnold diffusion "dominating" the dynamics!

 (1)   The author distinguishes between “resonance assisted tunneling (RAT)” and “chaos assisted tunneling (CAT)” but never explains the difference.  It would be useful to insert clearer definitions of these two tunneling mechanisms.  It is not clear if CAT actually exists in systems with 3 DoF (or more DoF).  CAT generally involves a symmetry that leads to the quantum tunneling.  Under what conditions do analogous symmetries exist in systems with 3 DoF?

I have put in a few more comments regarding RAT and CAT. A detailed discussion would lead to a rather large excursion from the main theme of the review. In any case, the references provided have ample discussion of the definitions of the two mechanisms. There is no  absolute or hard argument against existence of CAT in three or more DOF. As has been mentioned in the review, one can have several classically localised but disconnected regions on the web (at the same constant energy) that may get connected quantum mechanically on relatively faster timescales. Whether this mechanism bears all the usual signatures associated with a CAT process remains to be seen. 

  (2)   The author refers to “quantum dynamical localization”.  He should describe in more detail what this is.

It has been corrected to quantum localisation.

(3)     References 2, 51, 52, and 82 appear in footnotes, which puts them out of order and hard to find. Is this acceptable to the journal.

I have no comment on this -- it is a personal stylistic choice.

(4)     The author lists reviews of CAT but does include the recent Entropy journal review on CAT:  Entropy 2024, 26, 144.

Referenced in the revised version.

(5)     The English in the paper needs checking. For example, lines 88 and 173 in the proofs are garbled. 

Corrected to the best extent that I can.

(6)     The Figure reference on lines 277, 279, 283, and Fig. 6 is faulty.

Corrected.